# The Reuse of Biomass and Industrial Waste in Biocomposite Construction Materials for Decreasing Natural Resource Use and Mitigating the Environmental Impact of the Construction Industry: A Review

**DOI:** 10.3390/ma15124078

**Published:** 2022-06-08

**Authors:** Iwona Ryłko-Polak, Wojciech Komala, Andrzej Białowiec

**Affiliations:** 1Department of Applied Bioeconomy, Wrocław University of Environmental and Life Sciences, 37a Chełmońskiego Str., 51-630 Wrocław, Poland; iwona.rylko-polak@upwr.edu.pl; 2Selena Labs sp. z o.o., Pieszycka 1, 58-200 Dzierżoniów, Poland; wojciech.komala@selena.com

**Keywords:** mineral waste, bio-based waste, natural fiber, biomass, sulfur waste, copper flotation, fly ash, biochar, sustainable construction

## Abstract

The construction industry is the world’s largest emitter of greenhouse gases. The CO_2_ emission levels in the atmosphere are already reaching a tipping point and could cause severe climate change. An important element is the introduction of a technology that allows for the capture and sequencing of carbon dioxide levels, reducing both emissions and the carbon footprint from the production of Portland cement and cement-based building materials. The European Union has started work on the European Climate Law, establishing the European Green Deal program, which introduces the achievement of climate neutrality in the European Union countries. This includes a new policy of sustainable construction, the aim of which is to develop products with a closed life cycle through proper waste management. All efforts are being made to create generated waste and thus to support their production and/or use as substitutes for raw materials to produce biocomposites. This article reviews environmental issues and characterizes selected waste materials from the agri-food, mineral, and industrial sectors with specific properties that can be used as valuable secondary raw materials to produce traditional cements and biocomposite materials, while maintaining or improving their mechanical properties and applications.

## 1. Introduction

According to the Polish Central Statistical Office, in 2021, over 80,000 new residential buildings were commissioned, including single- and multi-family flats and buildings [1]. The construction sector is one of the most developed areas of the industry across the world. It is constantly growing and offers newer and innovative solutions for the end customer, thanks to which new technologies are developed and new components are sought [2]. The biggest challenge is to meet the requirements set out in the relevant standards and safety requirements, as well as the increasingly restrictive requirements implemented in recent years regarding energy efficiency and environmental issues. According to systems analysis, the main contradiction is that humanity wants to increase economic growth but does not want to live in the polluted environment caused by said economic growth, including the construction industry (Figure 1).

Another important aspect is adapting to new legal regulations regarding production, applications, and environmental aspects. The construction sector, including construction chemicals and their related characteristics and activities, strongly affect individual elements of the natural environment. Where water, energy, and natural resources are largely used, pollutants and other undesirable substances are emitted into the atmosphere, including greenhouse gases, and a significant amount of waste is generated [3]. According to a United Nations report [4], the construction sector is the largest emitter of greenhouse gases, reaching nearly 38% of global CO_2_ emissions. Carbon dioxide mainly comes from the production of cement. The cement industry is responsible for around 63% of the total CO_2_ emissions from the manufacturing process alone [5]. With these problems in mind, several plans have been developed to improve the environment and the economy. The first initiative was the United Nations Environment Program (UNEP) [6], which took steps to identify the environmental impact of buildings. The main focus is on the supply chain of building materials, the subject on which the report “Greening the Building Supply Chain” [7] was published, which addressed the environmental aspects of design and construction works and the production of construction products. According to the report, approximately one third of the world’s energy consumption is related to the use of residential and office buildings, and approximately 30–40% of gas emissions come from the construction sector, which, in turn, uses around three billion tons of raw materials, accounting for approximately 40–50% of global extraction levels. Many materials are obtained through the exploitation of non-renewable resources, which necessitates the use of recycled materials. In 2019, the European Union Commission adopted the name of the European Green Deal [7,8], the main goal of which is to create a modern, resource-efficient, and sustainable economy. It assumes net zero greenhouse gas emissions by 2050 and is expected to help with the COVID-19 pandemic [5]. In accordance with the Paris Agreement, which aims to reduce gases to zero, the European Union has launched the Fit for 55 program, in which it assumes that by 2030, these emissions will be reduced by 55%. Representatives of the cement industry [5] believe that it is necessary to decarbonize buildings, without which the intended goal will not be achieved. This process covers all stages of the construction of buildings, from designing buildings with materials with minimal or no carbon footprint using easily demountable, recycled, or renewable components for the sustainable construction and operation of buildings. This plan applies to newly constructed buildings, but the greatest challenge is to modernize the existing structures to meet the requirements of the Green Deal project [9]. In addition, as reported in [10,11], the prices of raw materials, building materials, and prefabricates increased significantly in the last year, according to an analysis of the Polish Economic Institute (PIE) [11]. The prices of building materials in the second half of 2021 increased by an average of 21.7% compared to in the previous year. According to PIE, such costs are increased by, among others, delays and reduced supplies of building materials from China, rising fuel and energy prices, and high wage costs. Another important problem is the availability of raw materials to produce building materials, where delivery times have been extended from a few to several weeks, or their access has been limited [12,13]. Considering the current economic situation and the restrictions related to the European Green Deal program, a good solution is to start using materials of natural origin that have a low carbon composition and do not absorb energy during their production, as well as the use of waste materials from various industrial sectors with interesting, unique properties [14].

## 2. Building Material Management—The Selection of Raw Materials and Production

In the application of lean management, the Theory of Solving Innovative Problems (TRIZ theory system) can be used [14]. This is an engineering approach for solving problems that, unlike other methods based on brainstorming, for example, uses the experience of designers, logic, and science to find a solution to a specific problem [15]. The construction industry is undoubtedly a very energy-intensive industry and simultaneously causes the greatest environmental pollution. Sustainable construction is a complex process that requires solutions to the rational use of natural resources and the reduction of environmental pollution. For this purpose, a good solution is the mentioned use of waste and/or recycled materials, designing new products that meet the requirements of construction materials standards.

Three groups can be distinguished in the TRIZ method:

**The supersystem**—A global-scale phenomenon and policy for managing and influencing the production of building materials.

**The system**—Available approaches and technologies for the production and application of construction materials in compliance with the requirements of standards, certifications, and people management.

**The subsystem**—Comprising techniques for the extraction and processing of raw materials and the production of building materials, which consists of the implementation of various products.

All of these groups are closely related and interdependent; the developed TRIZ method identifies problems resulting from the current economic and ecological situation and proposes possible solutions by applying a map of hypotheses.

The main problem areas at the level of the supersystem are defined below:

**DP1**—To develop the construction industry, humanity needs to excavate natural resources: non-renewable (e.g., water, minerals, and fossil fuels) and renewable (e.g., biomass and energy), causing environmental degradation and pollution.

**DP2**—To develop the construction industry, humanity needs to change the technology for sustainable construction, the transformation of production technology, and the use of renewable, waste, or recycled materials.

**DP3**—To develop the construction industry, humanity needs to develop new policies (e.g., European Green Deal) for the protection of non-renewable resources, the sustainable use of natural resources, zero waste, and the circular economy.

**DP4**—To develop the construction industry, humanity needs to change the technology for sustainable construction, searching for technologies and raw materials with low CO_2_ emissions and a small carbon footprint.

**DP5**—To develop the construction industry, humanity needs to implement a new policy of climate neutrality, i.e., the European Green Deal.

**DP6**—To develop the construction industry, humanity needs to increase their commitment to the use of waste and recycled materials for sustainable construction.

The main problem areas at the level of the system and subsystem are defined below:

**IdP1**—To develop the construction industry, humanity needs to invent and implement systems, technologies, and techniques for the minimization of the consumption of natural resources.

**IdP2**—To develop the construction industry, humanity needs to invent and implement systems, technologies, and techniques for the minimization of environment pollution, gas emissions, waste production, and energy consumption.

**IdP3**—To develop the construction industry, humanity needs to invent and implement systems, technologies, and techniques for the replacement of non-renewable resources by the sustainable use of natural renewable resources, including waste materials.

**IdP4**—To develop the construction industry, humanity needs to invent and implement systems, technologies, and techniques for decreasing the climate change impacts related to pollution from the construction industry.

**IdP5**—To develop the construction industry, humanity needs to invent and implement systems, technologies, and techniques for using renewable, waste, and recycled resources.

Analysis of the hypotheses map (Figure 2) indicated that there is a strong contradiction between the importance of environmental issues related to the construction industry and the production and application of building materials, meeting quality and safety standards, and the use of non-renewable resources combined with carbon fingerprints. The aim of this literature analysis was to show the possibilities of substituting the resources used in the construction industry with industrial waste and biomass, with a critical discussion of the strengths and weaknesses of these solutions. The option for the co-application of biomass and industrial waste to achieve a synergistic effect for the mitigation of weaknesses was also analyzed.

## 3. Characteristics of Waste

Waste: In general, in the European Union, according to the amendment to the Act on waste 2014/955/EU: Commission Decision of 18 December 2014 amending Decision 2000/532/EC on the list of waste pursuant to Directive 2008/98/EC of the European Parliament and of the Council [15], it is “a substance and/or an object which the holder concerned intends or is required to discard.” [15,16,17]. Waste management is understood as “collecting, transporting or processing waste, including sorting, together with the supervision of the aforementioned activities, as well as subsequent handling of waste disposal sites and activities performed as a waste seller or waste broker” [18]. On the contrary, according to the Act, waste management is “waste generation and management” [18], and material recovery is understood as “any recovery other than energy recovery and reprocessing into materials that can be used as fuels or other means of energy production; material recovery includes, preparation for re-use, recycling and earthworks” [18]. The classification of waste depends on the sources of its generation and the degree of nuisance or risk to health, human life, animals, or the environment. There are many divisions of waste, including substances of consumption or production, by their origin (consumption or production), degree of hazard, properties, and source [18]. In the European Union, 20 groups of waste are classified depending on their origin [18]. In recent years, in the construction industry, especially in the housing industry, a “low environmental impact” trend has become increasingly obvious, where there is a strong emphasis on materials of natural origin, as well as waste materials and recycled materials. These solutions are perceived as economical, energy-saving, and friendly to health and the environment. The unquestionable advantage of the above solutions is the use of waste materials [18,19]. The analysis of the hypotheses map (Figure 2) indicated another important contradiction: humanity wants to use new materials to save natural resources and to decrease the energy demand and impact on the environment, but there is concern about the properties of construction materials required to sustain economic growth (Figure 3). Therefore, this section of the article is an analysis of the current state of the art on the use of selected biowaste, biochar, and waste of mineral and industrial origin, with a description of the importance of their characteristics, important for their use in the production of biocomposite building materials and in construction, considering their environmental, economic, and strength standards. Therefore, a deeper and critical analysis of the application of numerous types of waste (Table 1) was carried out.

## 4. The Use of Biowaste in the Production of Building Materials

The agri-food sector, including agricultural and livestock farms and enterprises involved in the processing of paper, clothing, and other textile materials, as well as in the production and processing of food, generates various types of waste. The type and amount of this waste depends on the activity conducted in each farm. Waste management practices differ depending on the industry, and they can be subjected to various recovery and recycling processes, thanks to which we can obtain environmentally neutral materials with unique qualities.

### 4.1. Utilization of Lignin Waste

Lignin is a substance present in the cell walls of plants, corresponding, inter alia, to their stiffness and hardness [20,21]. It is an extremely complex polymer and is one of the components of biomass, filling every free space in plant cells [22]. Chemically, lignin is a natural polymer with a very complex three-dimensional structure, built from, among others, aromatic compounds, as well as from various phenyl alcohols [23]. Thus far, the main problems are obtaining pure lignin and its further transformation. Lignin is an unstable polymer that breaks down quickly during extraction attempts, so the main problem is its splitting and processing. This is why lignin is still quite a burdensome industrial waste for the environment that is of a relatively minimal importance, and only approximately 2% of its huge and still-growing resources, currently estimated at over 300 billion tons [24], are available for use. Lignin is mainly produced as biowaste in the production of paper after processing wood into paper pulp [25] and as a byproduct in the production of bioethanol from biomass [25]. In 2016, Li et al. [26] invented a way to obtain lignin from wood and break it down into its constituent substances. The main goal of the research was to develop two new catalysts that could convert lignin-forming compounds into useful chemicals and be used in products such as paints, insulating foams, and building mortars. Two catalysts were developed, the basic component of which was the addition of TiO_2_, and lignin was used as a carrier for the chemicals. The first catalyst additionally contained nanocomposites containing iron oxide Fe_2_O_3_, while the second one used zeolite (aluminosilicates) with a small addition of iron [27]. The publication describes laboratory tests in which lignin was used with the addition of the described catalysts and exposed to ultraviolet light. Both catalysts showed a high efficiency in the conversion of benzyl alcohol contained in the lignin structure into benzaldehyde, which is a substance used, among others, in in the production of pigments and other dyes [28]. After four hours, half of the original benzyl alcohol content had been converted. In industrial applications, a very important element is the selectivity of the reaction: if the reaction is more selective, its products are less contaminated with unnecessary and usually difficult-to-separate additives. As a result of the conducted reaction, the amount of the obtained reacted substance with the participation of the photocatalyst was approximately 90%. As noted by Li [26], all reactions in the model lignin, using the developed photocatalysts and under the conditions of natural light radiation, normal atmospheric pressure, and a temperature of 30 °C, proceeded automatically. Thanks to this, there is no need to maintain expensive and complicated infrastructure, as in the case of reactions in refineries. The obtained results are very promising, and further research on the effectiveness of photocatalysts in the processing of real, heterogeneous lignin with different compositions is still being conducted.

Since industrial lignin cannot be directly used to produce biomaterials, it is necessary to pre-treat it to reduce the sulfur content and improve its properties so that it can be used as a filler to reinforce composites and plasticizers. There is still ongoing research into technologies that can deliver high-quality lignin raw materials and derivatives, in addition to reducing greenhouse gas emissions, and can be used in sustainable construction [28]. Pandey [27] used two processes to isolate lignin from lignocellulosic raw materials: mechanical and chemical. In the mechanical process, cellulose and hemicellulose were removed by solubilization, leaving an insoluble residue, i.e., lignin. However, in the chemical process, the lignin was dissolved and removed, while the remaining polysaccharides constituted, in this case, an insoluble residue [21]. As a waste material, lignin may contain sulfur compounds that can affect subsequent applications, particularly sulfate lignin and lignosulfonates, which are used for pulping lignocellulose in industrial processes.

Another type of lignin is Kraft lignin [28], which contains approximately 1.5–3% of sulfur by weight, is soluble in an alkaline environment (pH > 10) and is characterized by hydrophobicity and a characteristic odor resulting from the presence of aliphatic thiol groups [29,30]. During the digestion process of lignin with sulfur, lignosulfonates with a highly cross-linked structure are formed, where the sulfur content is around 5%; along with two hydroxyl groups, namely, sulfonate and phenyl, they are soluble in the full pH range due to the sulfur contained in sulfate lignin, and lignin sulfonates are used as fuels to produce electricity and heat [31]. In addition to the group of sulfate lignin, there is also sulfur-free lignin, a sodium lignin characterized by a very high purity, low molecular weight, high content of silicates, and partial nitrogen [21], formed in the process of pulping. It is mainly used in the production of lignin from vegetable waste [28].

Yet another group of non-sulfur lignin is lignin extracted by the Organosolv method, which is obtained during the digestion of biomass from a mixture of inorganic or organic solvents in the presence of water [32]. The resulting lignin is characterized by a low molecular weight, high chemical purity, solubility in alkaline systems, and polar solvents [33]. The group of non-sulfur lignin also includes explosive lignin, characterized by a low molecular weight and solubility in organic solvents [34,35]. In addition to the lignin groups mentioned above, there are also ground lignin, pyrolytic lignin, and hydrolytic lignin. In addition to those mentioned in the study, there is lignin in solvent-free technology, where two types of lignin are developed—those extracted with ionic liquids and those with eutectic solvents [36]. Lignin is subject to modifications to obtain a greater scope of its application.

The main types of modification include [37]:Fragmentation or depolymerization, where its structure is broken down into aromatic monomers;Modification with the creation of new chemically active locations;Chemical modification of hydroxyl groups.


By modifying lignin, the hydrophobic materials for various applications, including as metal ion sorbents [38], polymer fillers [39], pharmaceuticals [40], and biosensors [41], as well as for selective extraction and its recovery, may be obtained [42]. Lignin and its derivatives are also used in the construction industry, mainly as an additive to cement composites, as one of the components of polyurethane foams and resins, and as a bitumen substitute to produce asphalt [42]. For the cement mixture to be workable without increasing the water content, so-called plasticizers [43] are introduced, thanks to which the mechanical strength and durability of the resulting composite may be increased [43].

Ouyang [40] described the action of substances added to concrete mortars. After the introduction of plasticizers (low-molecular-weight calcium lignosulfonates), an improvement in the physical and chemical properties of the cement mixture was observed. Additionally, it had the ability to adsorb cement particles on its surface and the ability to create foam. Lignosulfonates with a higher molecular weight showed an unfavorable air-entraining effect, which reduces the mechanical strength of the produced cement mortar [41,42]. Research has also been carried out on extracted pine lignin with formic acid, where it was fractionated by adding organic solvents [43]. As a result of the extraction, lignin of a high chemical purity with a low sulfur content and without nitrogen were obtained, while the fractionation resulted in obtaining two types of lignin, soluble and insoluble, differing in molecular weight, which, in turn, were oxidized with hydrogen peroxide, and the sulfonation process was carried out with the use of formaldehyde and sulfate sodium (IV). The resulting lignin was characterized by a strong improvement in the workability of the cement mixes compared to the extracted lignin [43].

Further results of lignin research were shared by Huang et al. [43], where they applied lignosulfonates–sulfomethylated alkaline lignin and enzymatically hydrolyzed lignin to cement slurry. The results of the test indicated a reduction in the water retention in the slurry and an improvement of the concrete compressive strength [44]. The use of Kraft lignin in concrete slurries was also investigated, and its influence on the obtained composite was examined, where it was subjected to radical polymerization at an early stage, resulting in a derivative of sulfate lignin with polyacrylamide elements. The resulting lignin lowered the plasticity of the cement paste, thus improving its workability, especially in the case of samples with the addition of kaolin and clinoptilolite (otherwise known as zeolite), where similar test results were obtained compared to commercial concrete admixtures [44]. Kalliola et al. [45] investigated the effect of O_2_ oxidized sodium lignin in an alkaline environment, which they used as a superplasticizer in concrete slurry. The tests showed that the lignin used is more effective than the commonly used lignosulfonates, showing greater plasticity and no air entrainment in the mixture, leading to a reduction in the compressive strength of concrete. Very interesting research was presented by Li et al. [46], where the authors used lignin modified with epichlorohydrin and diethanolamine (DML) for the production of Portland cement. The addition of DML improved the grinding and particle size distribution; in addition, when DML was introduced into the cement mortar, it influenced the mechanical properties during hydration and delayed the beginning and end of the setting time of the cement grout.

Klapiszewski et al. [47] developed new hybrid materials containing lignin. Two materials were created through mechanical synthesis, namely, sulfate lignin and magnesium lignosulfonate, which were used in concrete admixtures. The study used the FTIR method, which showed that the use of new materials improved the hydrogen interactions in organic and inorganic components, which changed the plasticity of the composite, and the mechanical strength of the building material also improved [48,49]. The next test was the use of the synthesized Kraft lignin–silica hybrid, which was added to the concrete mix in various doses (0.5% and 1%). After the tests, it was found that the admixture significantly affected the mix rheology, dispersion of cement components, low porosity, and setting of the final mechanical strength of the resulting composite [50].

Research has shown that lignin and its derivatives are materials with great potential, thanks to which it is possible to design hybrid systems that can be used in the design of cement composites with favorable rheological and mechanical properties. However, some evidence of a negative influence of lignin application on the workability and air-entraining properties has been identified.

### 4.2. Use of Hemp Fiber Waste

Hemp is a valuable raw material that is used in various industries, including construction [51].

For many years, hemp fibers have been one of the most important materials in the textile industry, produced as a result of the separation of straw, from which the shiver is also obtained. These materials are used to make various types of materials, such as clothing, headgear, tarpaulins, runners, and duffel bags. The main advantages of this material are its high mechanical resistance, smoothness, and natural fungicidal properties. It is also used to produce ropes and canvases, such as for sails [52]. In natural medicine and cosmetology, hemp products are used as a remedy for skin inflammation and skin and hair regeneration [52]. In pharmaceuticals, cannabis, including medical marijuana, is still controversial today. According to the latest scientific research, cannabis has many medicinal properties, but it should be remembered that marijuana is considered a drug. However, due to its health properties, work is underway on new forms of drugs, supplements, and product registrations all over the world [53].

The main source of hemp fibers is primarily straw, which undergoes a retting process [54]. The purpose of this process is to separate the bast fibers from the rest of the stem. “Retting” is the transition of straw to the fermentation phase through the interaction of organic substances. As a result of this process, pectin breaks down, acting as a “glue” connecting the woody part of the hemp with its fibers. At this stage, it is very important to regularly turn the straw over to ensure even distribution of the sticky substrates [55]. The fiber’s quality is influenced by its proper sprouting. The end of the process is determined by its color—the best time for the straw to turn dark gray. In addition, a specific, recognizable straw breaking sound is assessed, which also determines the quality of the fibers. Straws prepared in this way are formed into bundles and allowed to dry completely, then the fibers are extracted by breaking and crushing the stems, separating them, and cleaning the impurities. Then, the extracted fibers are prepared for further processing [56]. There are two types of hemp fibers: long, most often used in textiles and textile products, and short, made from the decay of long fibers and used as composite materials or insulation [55].

A novel aspect in the field of hemp applications is composites and biocomposites, based on binders and natural resins, which are reinforced with fibers of natural origin. The main advantages of biocomposites using lignocellulosic materials are their biodegradability, as well as their high mechanical strength, low specific weight, natural origin, and low acquisition costs. These composites are used in many branches of the economy, including in the construction industry, as well as in the automotive, aviation, and rail transport industries [57]. In construction, the use of both fibers and so-called hemp chaff can be found. Hemp fibers are used in the production of thermal insulation materials, the so-called hemp wool, bound with rice or corn starch. The addition of soda increases the material’s exposure to fire. The thermal conductivity coefficient of hemp materials is 0.04 W·m·^−1^ K^−1^ and is comparable to traditional thermal insulation materials. Due to the high content of cellulose (approximately 57–77%), these materials can regulate the level of humidity in rooms; such insulation can absorb moisture of up to 20% of its mass [58]. Hemp chaff, a byproduct made of lignocellulosic materials, is obtained from the processing of unretted hemp straw on a decorative line. A highly porous product, it ensures low thermal conductivity (approximately 0.082–0.144 W·m·^−1^ K^−1^) [58] and has a low density and good heat capacity, thanks to which it can accumulate heat [58]. In combination with lime, it is a lime-chipping composite that can be sprayed or form-worked to form the walls of a building, where the element is the wooden frame of the building. Depending on the density, it is characterized by high water absorption (98.5–150.5%); the higher its volumetric density, the greater the tightness of the composite and thus the lower the water absorption. The presented hemp composites have a very positive effect on the environment, as they have a high ability to absorb carbon dioxide in the photosynthesis process [58].

Building materials made of fibrous plant materials and lime binder are characterized by an alkaline environment, thanks to which they are resistant to the action and development of mold and fungi [57]. The addition of lime also ensures the materials’ resistance to fire—chaffs coated with a layer of lime binder show good fire-resistant properties. In 2016, for the first time, a house was built completely insulated with a lime–hemp composite [59]. Composites made of hemp are also characterized by their low production costs, very good insulation, thermal and ventilation conditions, high durability, and resistance to pests. They have vapor-permeable, antiallergic properties and are characterized by their low thermal conductivity [60]. The use of hemp fibers in the construction industry undoubtedly opens new opportunities. It is also a fully ecological method of using natural resources, which reduces environmental pollution, and companies are increasingly convinced of solutions using hemp [60].

### 4.3. Utilization of Bamboo Fiber Waste

Bamboo is one of the most widespread tropical plants in the world [61]. It is a plant with rapid growth, thanks to which it is possible to obtain material in a fairly short time. In China, efficient technology for producing bamboo fibers have been developed [62]. Bamboo fibers can be obtained in two ways:Mechanical extraction, where a bamboo fiber is obtained by means of steam under appropriate pressure in a mechanical press;Grinding.


The steam extraction method produces short fibers, characterized by their high strength, while longer fibers are obtained by roller pressing [63]. A chemical process through alkaline or acid hydrolysis removes the amorphous phases from raw bamboo. The chemical solutions (4% sodium hydroxide) used in this method affect the cellulose components of the fibers, removing approximately 40% of polysaccharides and lignin from bamboo fibers. We obtain short fibers using this method [63,64]. Bamboo was already known in ancient times, when the first houses were built from it; the interest in using this material is still growing, and several studies [65] have been carried out using this raw material to strengthen structures, replacing steel reinforcements. Nowadays, bamboo is used in the construction and furnishing of houses; moreover, furniture and accessories are made of it, during the production of which, waste in the form of fibers is produced. These fibers can be used to repair cracks in concrete, to reinforce concrete sleepers, or as an additive to reduce shrinkage in concrete mixtures [66]. The chemical composition of bamboo fibers is like the chemical composition of a tree; it contains cellulose, which is the main component of the fibers, hemicellulose, lignin, which is responsible for fibers’ strength, and water [67]. The cellulose contained in the fibers absorbs a large amount of water, which has a negative effect on said fibers. To minimize this, the fibers are rinsed with a solution of NaOH, KMnO_4_, and H_2_O [67]. Research has been carried out with the use of bamboo fibers in concrete [68], where the effect of fiber additions on the tensile strength, microcrack formation, and plasticity of concrete was observed. The test results were positive, where it was found that the bamboo fibers limited the shrinkage of the concrete, and the length of the cracks had a positive effect on the tensile and compressive strength. However, it had a negative effect on the workability of the concrete when using too much fiber waste; therefore, when designing the mixture, attention should be paid to the amount of fiber added to concrete [69].

Similar tests were carried out by Ende et al. [70], where self-compacting concrete with the addition of bamboo fibers mixed with lime flour was tested. Plasticity was assessed, and compressive and tensile strength tests were performed. The results showed that the addition of lime powder (approximately 10%) regulates the workability of a concrete mix, improving its plasticity and workability, as well as positively influencing the final results for tensile and compressive strength.

### 4.4. The Use of Recycled Fibers (Textile Fibers and Textiles)

The clothing sector and the related textile production use significant amounts of primary raw materials. This industry is one of the most environmentally damaging industrial sectors in the world. Reusing and/or recycling textile products allows for a significant reduction in the negative impact on the environment [71].

Textile production includes activities such as intensive cultivation and harvesting, soil degradation, and water scarcity. Various chemicals are used in the production of textiles, which has a negative impact on fauna and flora, as well as on human health. In addition, in recent years, a significant increase in CO_2_ emissions has also been noticed in this industry, which has given the textile industry the status of one of the most environmentally polluting [72]. Currently, cotton or polyester clothes dominate store shelves. Worldwide, it is estimated that over 60% of clothes can be reused, while the remaining 40% cannot be sold in their original form and are transformed or recycled into new products in many industries [73].

Textile waste is also present in modern construction, appearing in various forms, such as knitted fabrics, fabrics, or in the form of fibers. In residential construction, modern fiber-based composites have increasing potential as an alternative to traditional reinforcements of building structures. A textile fiber can be obtained by means of refining waste from the textile industry [74]. Textile fiber waste can come from municipal, commercial, or industrial sources, including household items (carpets, rugs, packaging, mattresses, and clothing). According to Morley [75], the largest share of textile waste (65%) from carpets was recorded in the United States; meanwhile, used clothes, bags, and shoes accounted for approximately 37% of the waste in England, while post-industrial waste accounted for approximately 68% of the waste in Australia [75]. The recycling process of post-consumer textile waste, in which the resulting product is fibers, is not the easiest one. It depends on the type of fiber being used. Textile waste is categorized and then sorted according to its color, type, and quality. It is examined using infrared or Raman spectroscopy, and then is subjected to a process of grinding into smaller pieces. Waste prepared in this way is transported to a rotary mill with embedded blades for breaking the material into threads and fibers [75]. Carpet waste requires preliminary detaching of the backing layer from the fibers with steam or water jets in such a way as not to damage the shape of the fibers [76]. The chemical recycling process is carried out by solvent extraction or with the use of deratized or non-deratized solvent. The entire process is carried out under a high temperature of up to 220 °C [76]. After filtering off the solids, the resulting material is cooled and precipitated into synthetic textile waste. In this process, depolymerization of a basic monomer (an oligomer can be used) allows for obtaining high-quality fibers, but it is quite an expensive procedure compared to the traditional method of obtaining fibers. Chemical recycling has a major disadvantage: As the process uses chemicals, some recognized as toxic or non-recyclable, the practical application of this process is limited [77].

Due to the properties of the fibers, they can be divided into two groups:Low-end fibers with a high tensile strength and a low modulus of elasticity (plastic);High-end fibers with a high tensile strength and a high modulus of elasticity (brittle).


Recycled low-end synthetic fibers such as nylon and polypropylene are widely used in reinforced concrete structures due to their good alkali resistance, water repellency, and excellent mechanical properties. According to Mohammadhosseini [78], concrete reinforced with textile trawls with lengths of 8–60 mm in the amount of 0.2–2% shows very good physical and mechanical properties. The addition of 20-mm-long polypropylene fibers derived from carpet waste in various amounts showed a reduction in the breaking strength of 38% for the addition of 0.25% fiber, 66.7% for 0.5% fiber, and almost 88% for 1.25% fiber, compared to a concrete mixture containing no fibers [78].

In the research of Mohammadhosseini [78], the compressive strength was also tested. The results showed that the use of recycled textile fibers causes a decrease in the compressive strength as the fiber content of the blend increases. There was a decrease of 21.3% in compressive strength with 1.25% fibers compared to a sample without fibers. These studies were, however, undermined by Wang [79], who showed that the addition of fibers in the amount of 0.08–0.16% with a length of 12 mm increased the strength of concrete to 40.12 MPa compared to concrete without fibers, where its strength oscillated at the level of 39.8 MPa. Similar results have been reported in the literature [80,81], describing the use of higher doses of fibers from textile waste, which leads to an increase in compressive strength. A concrete mix has also been tested for tensile strength; the addition of 1.5% fibers increased the tensile strength (2.9 MPa) by 31.8% compared to concrete without fibers. A similar effect was observed in the bending strength test: as much as 0.12% of the recycled fibers increased the bending strength by 27.2% (in the case of using 12- and 19-mm-long fibers, the results increased to 43.5%) compared to the samples without fibers. Improvements in the flexibility of the concrete mix were also noted [80,81]. In 2016, Barrera et al. [81] conducted a series of studies using textile materials, where they developed ready-made panels used inside buildings, composed of textile waste mixed with hydrated lime. The obtained material improved both the thermal and acoustic conditions of the buildings and contributed to a reduction in the impact of energy consumption related to the production of building materials and a reduction in greenhouse gas emissions.

In conclusion, the use of recycled textile fibers may cause a decrease in compressive strength; this parameter can be improved by using an appropriate number and length of fibers, thereby increasing the compressive strength while improving the bending strength.

## 5. The Use of Waste of Mineral Origins in the Production of Building Materials

Any mining activity, including the exploitation of deposits, leads to violations of the original state of balance of the natural environment; on the contrary, it contributes to intensive economic and social development [82,83,84]. Each exploitation is associated with the formation of byproducts of mining activities, which generate various types of heaps and landfills, constituting an unnatural element of the landscape, destroying the ecosystem, occupying the area in place of agricultural and forest crops, and having a negative impact on water and soil [85]. In Poland, according to the State Mining Authority (WUG), in 2019, ~68.6 Mt of mining waste was generated, which means an increase of 3.6% compared to the previous year [85]. Soon, according to the Supreme Court of the EU (EuGH) and EU Directive 2018/851 of 30 May 2018 [84] on the management of post-mining waste [84], including post-production mineral residues or mineral waste, the elimination and recycling of these materials will become very important. All of these activities lead to the necessity of using post-mining waste [85], possibly for the production of construction materials.

### 5.1. Reuse of Waste from Coal Mining

Coal waste is waste rock of Carboniferous sediments, among which there are coal seams. They can be divided into two groups:The first one is mining waste, generated in the course of mining works. It is mainly a waste rock in the form of large rock fragments.The second group consists of mining waste: waste rocks, which are deposited in the bottom and roof of coal seams, and overgrowths, which, during the exploitation of coal seams, get into the output and are extracted with the waste to the surface, then separated in the processing plant [86].


Depending on the machines and coal enrichment technology used, the tailings are of different granulations. Coarse-grained comprises fractions ranging from 20 to 200 mm and fine-grained ranging from 1 to 20 mm; flotation waste and other sludge have fine fractions from 0 to 1 mm [86]. The amount of generated mining and processing wastes depends mainly on the exploitation of the carbon series and layers, the geological conditions of the coal deposition, and the types of machines and technology used during coal mining and enrichment. The share of mining waste in the total mass is 15–18%, while that of processing waste is approximately 80% [86]. The main petrographic components of the described waste are loams, mudstones, sandstones, and gravel [86].

Loams are small pellitic rocks, ranging in color from light gray to almost black. Loafers lie mainly next to coal seams, where they form the ceilings and bottoms of said seams as well as the overgrowths in the coal, during the extraction of coal to the main mass of tailings. Their mineral composition is quite diverse, including, among others, illite, kaolinite, and small amounts of quartz, as well as ferric and carbonate minerals. This solid sedimentary rock, devoid of flaking, is a very valuable raw material for the ceramics industry and in the production of building materials [86,87].

Mudstones are fine-grained sedimentary crumb rocks formed due to the cementation of silt, and the main mineral components are, among others, quartz, feldspar, mica, and carbonates, and they may also contain crumbs of fine-grained rocks and clay minerals, phosphates, and iron compounds, which act as a binder of fine-grained materials [86,87]. Sandstones belong to the group of siliceous raw materials, the main component of which is minerals from the SiO_2_ group; there are also small amounts of feldspar and mica. Depending on the size of the grains, sandstones are divided into coarse, medium, and fine grains. Sandstones in the Carboniferous series are most often in the uniform form, sometimes multi-meter shoals; they also occur as inserts among mudstones and clay. They are occasionally found in tailings but are very common in mining tailings. Silicate raw materials are the main component of ceramic masses and glass sets. These raw materials are also the basic raw materials in the production of refractory materials, glass, and precious ceramics, as well as building materials and construction chemicals. They are used as the main component of cement mixtures, and they act as a chemically inert filler [86,87].

Gravel rocks are found sporadically, mainly in mining waste. They contain grains of quartz and feldspar, as well as crumbs of their parent rocks. These raw materials are widely used in road construction [88,89].

The largest group of waste produced and deposited in Poland is waste from bitumen coal mining and processing. They are mainly used in material engineering, construction, and road construction, as well as in the production of cement and construction ceramics. At least three quarters of the volume of concrete mixes consist of various types of aggregates, acting as a filler, and their main function depends on the size and shape. Fillers can work with cement, improving the packing of particles in concrete and making it plastic. Thanks to its well-packed structure, it can also replace some cement, thus not affecting the final mechanical strength [86,87].

### 5.2. Reuse of Waste from Copper Ore Flotation

The second waste producer, after coal mining, is the mining of non-ferrous metal ores. They constitute a finely ground waste rock containing residual amounts of useful minerals [89]. From the very beginning of operation, Polish copper mining has deposited 100% of flotation waste, and the possibilities for waste management are still being explored. As a result of coal output, approximately 94–96% of the mass of the extracted raw material is fine-grained flotation waste. The main characteristic of such waste is their very fine graining (less than 1 mm) and high humidity. After the dehydration process in filter presses, the water content is 20%. The content of the carbon substance varies and ranges from a few percent to almost 30%, and the sulfur content is usually above 1%. The main problem is their long-distance transport; in the wet state, this waste is characterized by a significant thixotropy, thanks to which it melts into a homogeneous mass, which makes it difficult to unload later [90]. The possibility of using post-flotation waste in the field of ceramics has been demonstrated using thermal methods. In these studies, in combination with other ingredients, brown and black pigments were obtained, which were later used in glazes such as ceramic glazes [90]. It was found that the addition of flotation waste to ceramic mixtures resulted in a reduction in the shrinkage of the produced ceramic products and a significant reduction in water absorption [90]. Equally interesting research results were observed with the use of flotation waste as a source of iron compounds to produce a hydraulic binder, i.e., Portland cement [91]. Depending on the chemical composition and nature of the flotation waste, it can be used in cement plants as one of the raw material components to produce Portland clinkers. The test results for Portland cement, produced with the addition of flotation waste, were compared with the cement produced from natural raw materials, showing that its properties are similar in both cases. These studies confirmed the possibility of using flotation waste as a marl substitute in raw meals to produce Portland cement. Flotation wastes can also be used as a major or minor component of cement, finding use as a cement setting regulator or as an alternative fuel in the production of Portland and alumina cement [90,91]. Miletic [92] described laboratory tests, the purpose of which was to determine the effect of the addition of flotation waste on the mechanical strength of paving stones. In addition to strength tests, parameters such as water absorption, abrasion (using the Böehme disc), and resistance to frost (frost resistance) were also tested. A concrete mix with various contents of flotation waste (5%, 10%, 15%, and 20%, respectively) was tested by weight. It was shown that the water absorption decreases with an increasing amount of flotation waste. However, the compressive strength tests of samples with the addition of floatation waste, conditioned for 28 days under laboratory conditions, showed a negative effect on the tested parameters. The compressive strength deteriorated as the amount of waste added increased. In the case of abrasion tests, the obtained results were similar, where the samples with the smallest amount of additive showed a slight loss in mass. It was also shown that the addition of flotation waste has a significant impact on the properties of concrete. Flotation waste has a positive effect on the improvement of concretes’ tightness and the reduction in porosity, although it unfortunately has a negative effect on mechanical strength and frost resistance. However, the remaining satisfactory results are an “incentive” for further research on the use of flotation wastes in the production of cement building materials [93].

### 5.3. Reuse of Ashes from the Incineration of Municipal Solid Waste and Biomass

It is estimated that the construction sector consumes ~14–50% of natural resources; thus, it is classified as the second source of carbon dioxide emissions in the atmosphere [92]. Due to environmental concerns, it is imperative to find alternative materials, which, in this case, may be waste materials. Municipal solid waste is defined as materials that are generated with human participation due to numerous activities [93]. One of the solutions in the field of waste management is the thermal conversion process, including incineration. The incineration of municipal solid waste and other waste streams produces a significant amount of fly ash residue, often containing valuable metals [94]. The fly ash from a combined heat and power plant that burns coal is probably the most-used pozzolanic waste material in the production of concrete. The first records on the use of ash for concrete dates back to 1980 [92,93,94]. Soto-Izquierdo [95] presented the results of a study of fly ash in terms of chemical composition; the size of the remaining powder particles was determined, and the ash was analyzed by means of scanning transmission electron microscopy (STEM) and energy dispersion X-ray spectroscopy (EDS). Then, the ash was added to the concrete mix in amounts of 5%, 10%, 15%, and 20%. During the analysis, the water–cement ratio was 0.7, and the aggregate–cement ratio (a/c) was 10. It was shown that the addition of 5% airborne ash to cement contributed to increases in the density of the mixture and improved the mechanical properties of concrete, thanks to the small size of the particles that filled the voids in the cement slurry. The possibility of using bottom and fly ash to produce cements and ceramics was also investigated. Pera et al. [96] used the possibility of replacing coarse aggregates (4–20 mm) in concrete with bottom ash from municipal waste incineration, in which they found crack formation and expansion when using “raw” ash. However, with the treatment of bottom ash with sodium hydroxide, the durability of the concrete improved, but the strength of this concrete decreased compared to the natural aggregate material. Lin et al. [97] melted fly ash from the incineration of municipal waste and replaced it with mortar cement; already using 10% bottom ash in the mixture improved the mechanical properties of the concrete. Ferraris et al. [98] used the method of vitrification of bottom ash at a temperature of 1450 °C without the use of admixtures. This method became the best method in solving the problems related to adding waste from municipal waste incineration plants to concrete.

Keppert et al. [99] examined ash from the incineration of municipal solid waste for an alternative to Portland cement and aggregates in concrete. Four different combustion ashes without prior treatment were analyzed in the research. The analysis covered bottom ash, fly ash collected from various boiler lines, and fly ash from electrostatic precipitators, and the compressive strength was measured. The test results showed that bottom and fly ash did not flow for the setting times of the prepared mixtures, while the ash from electrostatic precipitators delayed the setting by 24 h compared to the other samples. In the case of determining the flow, unfortunately, all of the ashes showed a decrease in the flow, which was the result of high-water absorption by the ashes; to keep the mixture consistency constant, the addition of plasticizers was necessary. Additionally, the research [95] showed that the fly ash from the boiler line showed pozzolanic activity, which positively influenced the compressive strength of the concrete cubes after 28 days. Fly ash from electrostatic precipitators showed the best test result; with its highest share, it decreased the compressive strength by only 18% compared to ash from the boiler line. Similar research results were obtained when replacing ash with aggregates. In these studies, the bottom ash showed a grain size distribution like that of natural aggregates. Fly ash showed greater fineness, disqualifying it as an aggregate substitute, unless it can be used as a microfiller [100]. The test results showed that bottom ash is the only one that has no negative impact on the strength and setting times, with the use of no more than 10%. According to Latz and Popławski [101], fly ash from biomass combustion can be used in concrete technology. Heavy metals, which are a problem in biomass ashes, are effectively immobilized by the components of hardened cement, and therefore do not pose a health risk to people and the environment. Research has also shown that biomass ash can be an active additive to a cement binder, thanks to which it increases the dynamics of the development of early strength for the cement composite and reduces its water absorption. The function of the target is to activate this resource using 20% of this additive. Biomass ashes can also be used to produce cementless binders and ecological high-ash composites.

### 5.4. Reuse of Slag Waste from Municipal Waste Incineration

As a result of the incineration of municipal waste, in addition to ash, waste such as slag is also generated. Both wastes can be used in construction as an alternative to mineral resources. Manufactured building materials with the use of waste slags may be characterized by their low mechanical strength; therefore, they are most often used in road construction [101]. According to Mądrawski [102], the contents of heavy metals, both in ashes and slags from municipal waste incineration, are relatively low, and undesirable substances possibly remain embedded in the concrete structure itself. In the process of preparing concrete with aggregates from incineration plants, attention should be paid to the reaction of cement with aluminum and zinc. This reaction causes concrete to swell and crack [103]. To avoid this undesirable phenomenon, slags and ashes should be properly seasoned and processed before production begins. For this purpose, raw materials are rinsed, washed, and treated with sodium hydroxide, and heavy metals are removed and subjected to a glass transition process. The processed and cleaned combustion product has appropriate chemical and physical properties that ensure the expected quality of the concrete [104]. However, it should be remembered that the properties and chemical composition of combustion products depends directly on the properties of the waste to be burned and the location and waste management regulations in force in a specific region, and will be different; therefore, the concrete produced from them will have different characteristics. Mądrawski [102] tested slag from a waste incineration plant in Poznań, Poland, from which concrete samples were made, and the pozzolanic activity was determined by grinding the slag into dust and using it to prepare mortar. After 28 days, the conditioned samples were subjected to compressive strength tests and compared to samples mixed with silica dust and fly ash. The additives were dosed in the same volume proportions. As a result of the analysis, the unfavorable swelling phenomenon caused by the presence of aluminum was confirmed. The obtained strength results were comparable, thanks to which they could be further used in the production of building materials [101].

### 5.5. Reuse of Sulfur Waste

Sulfur waste is another waste from mining and industrial activities. Some of the wastes from mining activities are sulfur and a byproduct of sulfur production, the so-called sulfur cone. In road construction, it is used as an additive to bituminous masses. Sulfur cones are a filter waste from the processing of copper ores in the form of gray lumps. In terms of physical and chemical properties, they are a conglomerate of sulfur and minerals and contain limestone with admixtures of clay and gypsum and traces of strontium sulfate, barium, iron, and aluminum oxide. Sulfur cones are a material with a diversified grain fraction, and a soft material with low abrasion, low compressive strength, and medium water absorption [103]. Sulfur waste can be successfully used in construction, e.g., in the production of concrete composites. Analyses of material recycling in construction pay attention to the so-called carbon footprint of a construction product, i.e., the energy consumption of the production process. Due to cement production technology, this process is classified as very energy-consuming. The cement firing temperature is 1450 °C, which has a negative impact on both the carbon balance and the environment. Sulfur polymers can be an alternative to cement production, where the energy consumption of sulfur polymerization is much lower than that of cement [104]. So-called sulfur concrete can become a type of concrete used in both the road and construction industries. The production process of concrete from sulfur allows for the recovery of stored sulfur waste materials. The binding of sulfur concretes is a physical process, so no chemical reaction takes place here. To make such concrete, it is necessary to first create an environment in which workability of the mixture is possible and changes in the crystallographic system take place [103]. The main problem identified by Bahrami [105], Cholerzyński [106], and Halbiniak [107] is obtaining the temperature of the sulfur binder, which should be between 130 and 140 °C. This is the temperature at which sulfur becomes workable, i.e., sulfur changes into liquid form. It should be added that sulfur is a flammable material, meaning spontaneous combustion occurs above 170 °C, which is a great difficulty in such a process [106]. The advantage of sulfur concrete is its thermoplastic properties; it is like asphalt concrete, in that the temperature of processing is lower than that of asphalt concrete, so it is not deformed in the process of road use during high temperatures in the summer. Thus, there are two main directions for the use of waste sulfur polymers—the first is as an asphalt binder and the second is in the replacement of hydraulic cements [107].

Helbrych [108] examined the effect of the polymer from sulfur industrial waste on selected concrete parameters and determined the appropriate level of sulfur content in concrete composites. Sulfur polymers, waste from the purification process of copper and other non-ferrous metals, which were modified with styrene in the amount of 5% of their own weight, were used for the tests. A quartz aggregate with a grain size of 2–8 mm and white quartz flour with a grain thickness of 0.065 mm were used in the tests. Three types of sulfur concrete samples were made with sulfur polymer contents of 20%, 30%, and 40%. The test results confirmed the positive effect of the use of sulfur polymers obtained from industrial waste in the amount of 30%, where high and early compressive and bending strengths were obtained. The tests for the frost resistance of the samples showed no cracks, and the total mass of the concrete defects, where damage to the corners and edges was observed, did not exceed 5% of the sample mass. Compared to the non-frozen control samples, the decrease in compressive strength after freeze/thaw cycles did not exceed 20%, but it came very close to this limit. The conducted research has shown that it is possible to successfully use polymers from the recycling of sulfur materials in concrete composites. The obtained sulfur concrete is characterized by a very fast setting time, and the obtained results confirm that it can be an alternative for quick repair or construction works onsite, based on the conditions necessary for the preparation of this type of mixtures onsite. Książek investigated cement composites impregnated with polymerized sulfur waste [109]. As part of the preparation and production of specially polymerized sulfur, sulfur waste was first melted and then polymerized to a temperature of 150–155 °C. For comparison, a second sample was additionally made, where previously heated carbon black waste with a granulation of 0.330–0.990 μm was added to the composition of specially polymerized sulfur in the amount of 2–4%. During the analysis of the effect of impregnation of specially polymerized sulfur, the flexural strength was tested, and the surfaces of their fractures were observed. For this purpose, a scanning electron microscope was used, aiming to reveal possible defects arising during impregnation. The results of the analysis showed that the impregnated surface of the cement composites had a very tight structure and was devoid of open pores (capillaries), air voids, microcracks, and other defects, regardless of whether carbon black waste was added or not. Impregnation with a specially polymerized sulfur protects concrete surfaces against the aggressive effects of the environment, which reduces the corrosion of such materials and seals the surfaces well, while also strengthening it. All of the conducted studies show that it is worthwhile to conduct further research on the use and application of sulfur waste in the construction industry.

## 6. The Use of Biochar

In recent years, there has been strong interest in the technology for the thermochemical processing of biomass, biodegradable waste, and sewage sludge into biochar [110]. The issue of the potential use of biochar itself in various industrial sectors, such as energy, agriculture, environmental protection, pharmaceuticals, textiles, and construction, has been discussed [111]. Biochar in cement systems is a new area of research aimed at improving some properties of the final product, establishing protection against hazardous environmental elements such as corrosion, and using it in the sequestration of carbon in cement materials as an alternative method to the carbonization of cement materials [112]. Research has been carried out, according to which biochar as an admixture to cement products shows such parameters as high chemical stability and reduced flammability [112], increases its effectiveness as a hardener [113], and allows for the possibility of carbon capture and storage [113]. Biochar is a material of biological origin with properties like charcoal. It is obtained mainly from biomass residues resulting from agricultural waste, e.g., manure, poultry production, or residues from pressing oil, agri-food processing, sawmill production, and forestry production. It can also be obtained through segregated biodegradable municipal waste or sewage sludge [114,115]. Biochar may be produced in the process of biomass torrefaction at a temperature of approximately 200–300 °C or under pyrolysis conditions of 400–750 °C, under close-to-natural pressure. It mainly takes place in the transformation of hemicelluloses, or in the transformation of cellulose and lignin. The result of this process is a 30% weight loss, the release of so-called torgas, and the obtaining of biochar [116,117]. Biochar has very interesting properties, such as the ability to bind organic and inorganic pollutants and the ability to retain nutrients and water—the raw material can be used as a soil improver or a sorbent to remove pollutants [118]. Biochar is used in the production of animal feed and silage, and can also be used in construction as an insulating material, as a humidity regulator, or in energy storage in capacitors, as well as in many areas of life [119,120]. The use of biochar is primarily determined by its physical and chemical properties, which depend on the substrate used and the conditions of the pyrolysis process [121]. In its structure, biochar shows a high degree of chemical and microbiological stability.

The important physical and chemical parameters include [121]:Chemical composition (depending on the substrate and the pyrolysis process);Stability (low susceptibility to degradation and microbiological decomposition);pH (neutral or alkaline);The content of micro- and macroelements (including calcium, phosphorus, and magnesium);Micropollutants such as heavy metal ions, dioxins, and polycyclic aromatic hydrocarbons (PAHs);Low thermal conductivity—the ability to absorb water (up to 5 times higher than its specific weight).


Marris [122] indicated that a higher temperature in the pyrolysis process causes a lower yield of the obtained biochar. On the contrary, a high temperature may contribute to optimization of the aromatic structure, which increases the durability of biochar, its specific surface area (higher electrical conductivity), and its porosity [123]. The latter two parameters may influence the adsorption of heavy metals. Biochar may have a different specific surface as well as internal pores of different sizes, which depend on the type of raw material used and the pyrolysis process itself and may also create easy access to metal ions from the environment [124]. Meanwhile, the surface area and porosity of the feedstock have a lower influence on the adsorption of metal ions than the functional groups that contain oxygen. They are responsible for the high sorption of Pb on low-temperature biocarbon (250 and 400 °C). On the contrary, intramolecular diffusion is responsible for the low adsorption of Pb on biocarbon obtained at a high temperature (500 and 600 °C) [124]. In the process of the sorption of metal ions, the pH value is very important, which varies depending on the type of metal. The pH of the solution affects the speciation of the metal and the surface charge of the biochar, and its change affects the comprehensive behavior of the functional groups (carboxyl, hydroxyl, and amine) [116,117,118,119,120,121,122,123,124]. In recent years, the interest in biocarbon as a filler in the production of various composites [125,126,127,128], due to its large specific surface, hydrophobicity, and high carbon content, can improve the mechanical and physicochemical properties of these composites. Gupta et al. [115] investigated the addition of biochar as a filler in biocomposites and found that it reduces the biodegradable polymer and the cost of its production, while improving the mechanical properties of the composite itself. Restuccia et al. [129] made two cement composites with different percentages (from 0.05% to 1%) of two biochars made of different biomasses: coffee and nut shells. A superplasticizer was also added to improve their workability. Each sample made was tested for its bending and compressive strength, and elemental analysis using XRF was performed to characterize the properties of the produced biocomposites. The test results showed that the addition of both biomasses had a positive effect on the mechanical properties of the biocomposite, increasing the bending and compressive strength of the samples with the addition of as little as 0.5% of biochar compared to reference samples without biochar. The results of the XRF analysis showed that, depending on the biomass used to produce the biochar, the chemical composition will be different. As a result of the research [130], the biomass from nut shells showed a high percentage of carbon content and was characterized by high chemical purity, while the biochar produced from coffee powder showed the formation of potassium and calcium salts, which may contribute to blooming on the surface of the composite.

Due to the properties of biochar related to the removal of toxic metal ions from the environment, namely, a low thermal conductivity and natural sorption capacity, this raw material is suitable for use in cement building materials [131]. The most durable groups in the construction industry include all types of masonry structures. During a long period of operation (150–200 years), these structures are subject to degradation by various physical–chemical, biological, and mechanical factors, as well as by neglect. To restore them to their former glory, renovation is carried out. The greatest threats to a building’s structure are condensation moisture, which leads to the development of toxic mold fungi, and capillary moisture, which is the basis for the precipitation of destructive salts [131]. With prolonged moisture, there are significant losses in the construction and finishing layers. Many works look for the causes of these damages, referring to the incorrect use of lime, cement, and cement–lime mortars, which are not suitable for the repair and renovation of salty and damp walls. In the literature on renovation works and complementary mortars, the use of biochar as the main component of a cement mixture has been described [130]. By referring to the properties described above, biochar can be used to insulate a building, thus regulating the humidity in historical buildings. Tokarski and Ickiewicz [131] indicated the possibility of using biochar in the renovation works of buildings by making a plaster mortar in which lime, cement, and biochar were used in the proportion of 50% of the sand volume. The material used for the tests differed from one another in terms of the formula, and the carbon content in the biomass was 70%, 80%, and 90%. After the analysis of the preliminary research on the plaster structure, a very good thermal insulation and good influence on the indoor microclimate were demonstrated. Among others, the thermal properties of the mortar with biocarbon were investigated, carrying out tests of the thermal conductivity coefficient λ of the designed supplementary mortars [116,131]. The test results were positive, showing a value for the thermal conductivity coefficient three times lower than traditional lime–cement plaster. The work of Tokarski and Ickiewicz [132] also referred to strength tests, where samples with the use of biochar showed a high mechanical strength and positive results in terms of the bending strength, but insufficient results regarding compressive strength. Nevertheless, this leaves room for further work in this area. The lathes were also tested for physical properties, such as the determination of fresh mortar—all tests were positive; moreover, the absorbability of the mortars and the capillary rise were tested, and depending on the amount of biochar used, the samples obtained the required value. According to Tokarski [131], plasters based on biochar contribute to the regulation of room humidity (in the range of 45–70% of optimal humidity); they have anti-drying and air-purifying properties, exhibit thermal properties, bind toxins, are fungicidal, and protect against radiation.

Research has also been carried out on the use of biochar for concrete reinforced with polypropylene (PP) fibers. Their use is one of the best known and most used methods for the microreinforcement of concrete [132]. There are several studies [130] that have shown that fibers can weaken mechanical strength and increase permeability. In this case, it can be effective to use biochar as a coating on fibers. The use of biochar with polypropylene fibers may contribute to the sequencing of large amounts of carbon in building materials, thereby reducing CO_2_ emissions released into the atmosphere [130]. For the last few years, the interest in this material has been constantly growing [130], and research has been carried out [130] in which the addition of 1% biochar by the weight of cement significantly increased the fracture modulus compared to tests without the additive. As a result of the research by Choi et al. [130], the effect of polypropylene fibers covered with fresh biocarbon and CO_2_-saturated biochar and their use in cement mixtures were demonstrated, as were the permeability and the study of mechanical properties. The addition of fibers coated with fresh biocarbon slightly decreased the flow of the blend, but significantly increased the final mechanical strength and permeability compared to the control sample without fibers. On the contrary, the presence of CO_2_-saturated biochar caused carbonation of the cement mortar, which decreased the final mechanical strength [130]. These research results should be an incentive for further research on the use of biochar and its forms in mortars or cement mixtures. It is an alternative to building materials for energy storage and property improvement. Currently, research is being conducted on the regulation of pyrolysis conditions, the development of biocarbon aggregates due to its porosity, the development of nanobiochar, which will open up the possibility of replacing some Portland cement with biochar, and the possibility of using it to improve carbon dioxide sequestration in cement materials [133]. Based on these data, biochar can be used not only in the design of plasters, but also other cement materials, such as insulation mortars, tile adhesives, masonry mortars, and floor mortars.

## 7. Synergy of the Simultaneous Addition of Biowaste and Mineral Waste

The production process of ordinary Portland cement and the production of building materials and all concrete structures are the main factors in the emission of greenhouse gases into the atmosphere, thus creating several unfavorable problems. Therefore, there is a need—or even a necessity—to introduce sustainable construction and thus to produce an alternative ecologically green concrete. For this purpose, the use of waste materials from various industrial sectors, biowaste, and recycled waste, which can be successfully used as replacement or complementary materials in the production of building materials, such as mortar or concrete, can be an excellent solution.

Many natural resources are used in the production of biopolymer composites used in various applications. The use of waste from various industrial sectors as a filler or additive, or the replacement of commonly used raw materials with them, provides great economic benefit. The use of more than two raw materials in one component forms hybrid composites [133], in which the reinforcement is, for example, two types of fibers or two different fillers. Such composites provide many benefits where both natural and industrial waste are used and where they are minimized and used in an effective way, while providing materials with strong mechanical properties.

On the basis of the analysis of waste materials, several good solutions can be identified:In order to prevent the excessive use of natural resources in the production of construction products such as concrete or mortars, we can successfully use mineral waste. With the removal of impurities and the selection of granulation, they can replace natural fillers without affecting the final properties of construction products.In cement production, flotation wastes can be used as a substitute for margal in raw meal for Portland cement production, and flotation wastes can be used as major or minor components or as an alternative fuel. In the production of cement itself, part of it can be replaced with fly ash from the incineration of municipal waste, where, in this process, the fly ash reacts with the calcium hydroxide of the cement, finally giving much more durable and stronger compounds. Additionally, fly ash reduces CO_2_ emissions and energy consumption. Sulfur waste and biochar work similarly. As already mentioned, the production of cement is a very energy-intensive process, and it is responsible for the majority of carbon dioxide (CO_2_) emissions released into the atmosphere, while the production of biochar reduces CO_2_ emissions released into the atmosphere. Replacing some of the cement with the described waste is a beneficial alternative to reducing CO_2_ emissions and energy consumption, while maintaining the standard parameters of the cement produced.The workability of concrete and cement mixes: The addition of natural fibers, such as hemp, bamboo, or recycled textile fibers, as well as the addition of biomass fly ashes, adversely affect the plasticity of the mixes, thus lowering the mix viscosity. To prevent this, it is necessary to use plasticizers, and a solution may be the use of lignin derivatives of waste raw materials, which have a positive effect on the application properties while contributing to the reduction of CO_2_ emissions. The addition of biochar also reduces the decrease in the viscosity of the mixture, thus favorably reducing the carbon footprint in the finished biocomposite.The mechanical strength of concrete and cement mixtures with the use of precipitation: The addition of waste such as fly ash from biomass combustion, sulfur waste, biochar, and natural fibers positively influences the final mechanical strength.The tightness and frost resistance of concrete mixes: The addition of copper flotation waste positively improves these properties, but adversely affects the mechanical strength of the concrete mix. The addition of ash, sulfur, or biochar may improve this parameter.


Industrial waste has many applications. It is produced in very large amounts and is often unused, which is a waste of resources that can be used in the production of building materials, providing products with favorable properties regarding both the obtained composite and the environmental impact. The conducted analysis should encourage further research in this area, where it is possible to find effective and innovative solutions and ways of using waste.

## 8. Summary

The construction industry undoubtedly has a very large impact on energy consumption and CO_2_ emissions and a negative impact on the environment. A good solution is to use waste materials that can be used as a partial replacement for cement and/or aggregates or as fillers or additives to cement mixtures. This will not only reduce energy consumption in the development of new products, but also increase their life cycle. The construction industry has enormous potential to use waste from agricultural, mining, and industrial activities in its products. Many designers and contractors are aware of the unique properties of such products, which can significantly affect the energy consumption of buildings. Therefore, new technologies and materials, especially waste, for the production of construction materials are being searched for and developed. Biomass-origin waste containing fibers may be used as a component of construction materials, replacing part of the aggregate, increasing their mechanical properties such as frost resistance and their compressive strength properties, and reducing their CO_2_ emissions; however, it negatively affects the rheological properties of cement mixtures, which may cause some issues with meeting the construction materials standards. Mineral waste may be used as construction materials; its use can replace both the aggregate and the binder, reduce the extraction of natural resources, and improve the properties of cement, while reducing greenhouse gas emissions, but overdosing of these wastes may lead to a decrease in the mechanical properties and workability of the cement mixture. Another synergistic solution is the use of biochar from biomass and/or biowaste: it improves mechanical properties, has a positive effect on CO_2_ emissions, reduces the carbon footprint, and alleviates the problems associated with the use of untreated biowaste fibers and the identified problems related to the use of mineral waste.

The contradiction is that mankind wants to use new materials to conserve natural resources and reduce energy demand and environmental impacts but building materials that meet the required properties to sustain economic growth must be produced (Figure 4).

The use of organic waste, including biochar, will result in biological potential, thanks to which the construction industry can become a permanent part of sustainable green construction. The most important measures to this end include the minimization of energy consumption in buildings, the prudent use of natural resources, tighter control of the emissions of harmful substances, and reductions in the industry’s carbon footprint (Figure 5).

Reuse of both mineral and natural waste may limit natural resource excavation and result in the implementation of circular-economy approaches (Figure 6). These guidelines should be used when selecting materials for construction. The main approaches in the construction industry should be the use of renewable energy resources for raw material extraction and processing and the greater use of recycled waste, which will further increase its potential. These actions will permanently become part of a sustainable development policy.

## Figures and Tables

**Figure 1 materials-15-04078-f001:**
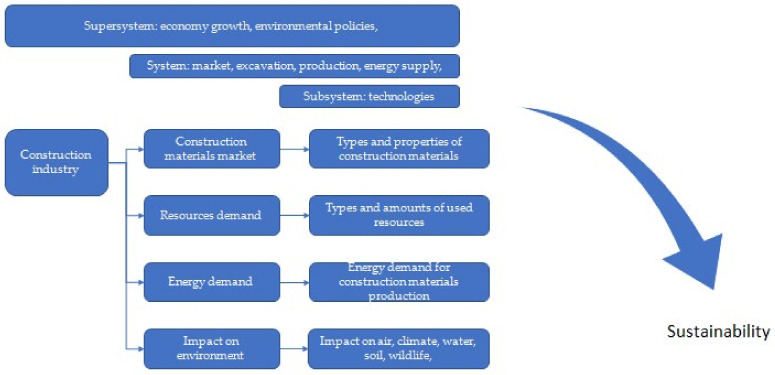
The current supersystem for producing building materials and the problem of environmental pollution.

**Figure 2 materials-15-04078-f002:**
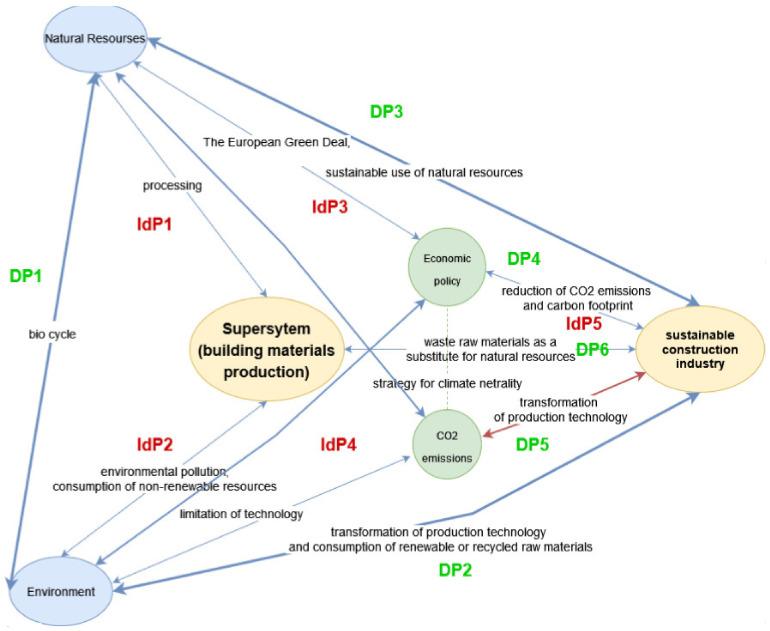
Map of hypotheses concerning the ecological and economic problems related to the construction industry.

**Figure 3 materials-15-04078-f003:**
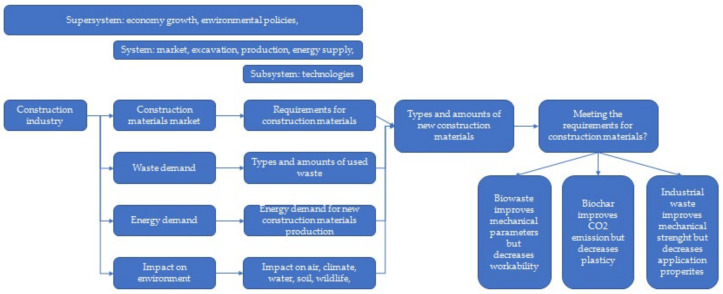
The proposed supersystem for producing building materials using waste, with an indication of the problems resulting from quality standards.

**Figure 4 materials-15-04078-f004:**
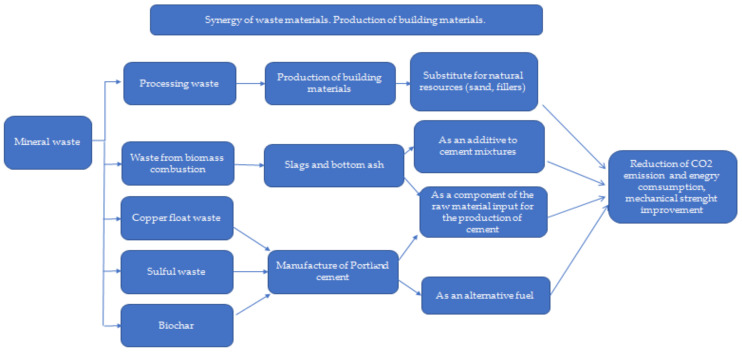
Synergy of mineral waste in chemical constructions.

**Figure 5 materials-15-04078-f005:**
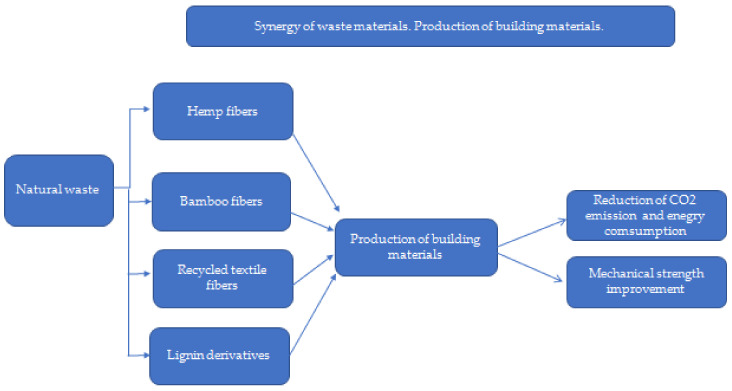
Synergy of natural waste in chemical constructions.

**Figure 6 materials-15-04078-f006:**
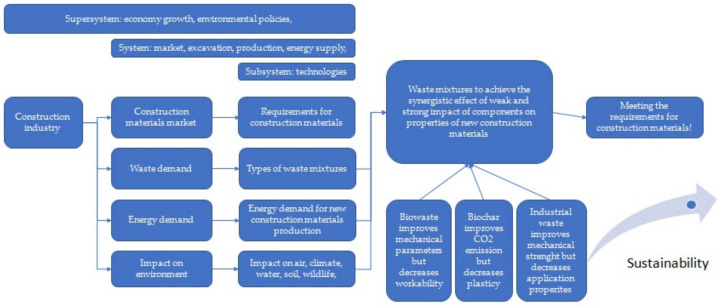
The proposed supersystem for producing building materials using mixtures of waste to meet the quality standard sustainability principles.

**Table 1 materials-15-04078-t001:** A list of the analyzed waste utilized as components of construction materials.

Waste Group Code according to 2014/955/EU: Commission Decision of 18 December 2014 Amending Decision 2000/532/EC on the List of Waste Pursuant to Directive 2008/98/EC of the European Parliament and of the Council	Waste Group Name	Reference
01 (01 01; 01 02; 01 03; 01 04; 01 05)	Coal waste, waste from the extraction of copper ores and other minerals.	[18]
02	Waste from the agricultural sector, horticulture, plant production.	[18]
03	Waste from the wood processing sector and the production of panels and furniture, and from paper processing, including pulp and cardboard.	[18]
04	Waste from the textile industry.	[18]
06 06	Waste from the chemical processes of sulfur production and processing and desulfurization processes.	[18]
10	Waste from thermal processes.	[18]

## Data Availability

Not applicable.

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
