# Peer review of "The Reuse of Biomass and Industrial Waste in Biocomposite Construction Materials for Decreasing Natural Resource Use and Mitigating the Environmental Impact of the Construction Industry: A Review"

_materials, 2022, doi:10.3390/ma15124078_

Round 1
Reviewer 1 Report
The manuscript is a literature review that analyzes selected waste materials from various sectors of the economy that can be used as additives or partial substitutes for natural resources in the production of cement and cement building materials. The manuscript gives more attention to the material itself and less to its influence on building materials. Anyway, it provides the reader with a vision of how to select and apply these wastes in concrete. Based on this, I suggest the authors review the title of the manuscript to make it fit the current content.
- The quality of the figures needs to be improved
- The abstract should be enhanced to highlight key findings
- Line 18, delete ‘and in’
- The reference [94] was lost, please check
- In introduction, it doesn't point out why this review is done, why the synergy effect between biomass and industrial waste in construction materials is considered? Is there anything special about biomass as compared to other industrial wastes
- What is your Section 4?
- In Section 3, the authors have given the definition and source of waste, so what are its characteristics? Physical and chemical?
- When the authors describe the modification of cement concrete by biomass material, please quantify these performance improvements and indicate by how much or from what to what
- It is desirable for the authors to make a table to compare the performance improvement of cement concrete by different biomass materials. In this way, readers can clearly understand the improvement of different performance of concrete by different biomass materials. This recommendation also applies to mineral waste.
- In Section 7, the authors propose synergies between bio-waste and mineral wastes. Is there any literature to show these
- Based on the content of this manuscript, I suggest that the authors remove synergistic effect from the title
Author Response
The responses to the Reviewer's comments are in the attached file.

Reviewer 2 Report
Dear Authors,
The recycling of industrial waste is a global problem and the authors have managed to capture the essential elements highlighting through this paper the latest achievements in this very current field.
As a week elements
The abstract could be more consistent because the paper contains enough things that can be highlighted and that should arouse the author's curiosity. Looks like you didn't make it in a hundred words.
Figures 1, 3 and 4 need to be remade because they are not clear.
The texts in sections 2, 3, 4 and 5 must be formatted according to the guide (justify).
Line 86 in parentheses is a vertical bar that must be deleted.
Some materials could be exemplified with the related pictures to be more suggestive even if they belong to other authors.
And in the term of notable elements
The paper proposed by the authors is very topical and very well documented through the 147 bibliographical references.
Author Response

(The authors gave the same response as above.)

Reviewer 3 Report
The paper presents an interesting review, or at least is attempting to, on the synergistic effect of biomass and industrial waste reuse as a good measure to slow down the use of raw materials in the construction industry.
The manuscript seems a bit disorganized with lots of information that is presented and without any clear, straight forward, connection from one topic to another. The sentences are too long and difficult to understand. It is highly recommended to use shorter sentences in order to clearly convey the message to the reader.
Please find below a series of suggestions the authors may consider in improving the manuscript.
Lines 86, 202, 422, 952, 957 - try to avoid writing in the the personal form since this is a scientific paper. Replace "we can use..." by "The theory... can be used (line 86)".
Line 41 - what are the "other characteristics" the authors refer to?
Line 59 - In 2019 EU adopted the Green Deal Act (not will act!)
Line 75 - please remove "the previous one"
Line 79 - "or there is no such thing"? What exactly do the authors mean by this statement?
Line 161 - please remove the double typing of "or is required to discard"
Line 196 - where is Section 4 starting?
Line 206 - "to for their coherence"?
Line 209 - what do you mean by "obtain energy of higher density"?
Line 218 - why is reference 27 cited before reference 26? Please cite the references in the order of their appearance in the manuscript. Similar comment for line 423, reference 99 and line 435, reference 71
Reference 30 is not cited in the text
Line 291 - what "substances" are the authors referring to? "Concrete mortar" or "cement mortar"?
Line 343 - the statement can be removed entirely as it has no connection to the topic at hand
Line 349 - reindeer?
Line 350 - try replacing "subtlety" by "smoothness" as I think it is closer to the intended meaning
Line 425 - did you mean "replacing the steel reinforcement"?
Line 468 - reference 73-76 need renumbering
Lines 548-551 - the meaning of the paragraph is not clear, please rephrase.
Line 363 - what is the meaning of the sentence?
Lines 576-581 - this paragraph can be entirely removed.
Line 630 - tested by weight?
Line 669-671 - shouldn't melted fly ash replace the cement mortar?
Line 889 - who is "he" describing the use of biochar?
Line 916 - how can something "sequence large amounts of carbon"?
Section 7 - which should be the main focus of the paper, at least according to the title, does not provide any state of the art review. The authors use only general statements without any citations of scientific works.
In my opinion, the previous sections should be reduced to the bare minimum so that the reader could get an overview of the current state of the art. The main focus, with lots of comments and conclusions drawn from the scientific literature should be in section 7.
The reference list does not seem to follow the guidelines of the journal. The authors are highly encouraged to use a reference management software / plug-in such as Mendeley, EndNote or any other in order to make sure the references are correctly cited in the manuscript and that the template is followed.
Author Response

(The authors gave the same response as above.)

Round 2
Reviewer 1 Report
I am satisfied with the author's reply and the article can be accepted and published.
Reviewer 3 Report
The authors did a very good job at reorganizing the paper and emphasizing the synergistic effect of bio and mineral wastes.